# Stochastic palmitoylation of accessible cysteines in membrane proteins revealed by native mass spectrometry

Remco N.P. Rodenburg[1], Joost Snijder[2], Michiel van de Waterbeemd[2], Arie Schouten[1], Joke Granneman[1], Albert J.R. Heck [2] & Piet Gros[1]

Palmitoylation affects membrane partitioning, trafficking and activities of membrane proteins. However, how specificity of palmitoylation and multiple palmitoylations in membrane proteins are determined is not well understood. Here, we profile palmitoylation states of three human claudins, human CD20 and cysteine-engineered prokaryotic KcsA and bacteriorhodopsin by native mass spectrometry. Cysteine scanning of claudin-3, KcsA, and bacteriorhodopsin shows that palmitoylation is independent of a sequence motif. Palmitoylations are observed for cysteines exposed on the protein surface and situated up to 8 Å into the inner leaflet of the membrane. Palmitoylation on multiple sites in claudin-3 and CD20 occurs stochastically, giving rise to a distribution of palmitoylated membrane-protein isoforms. Non-native sites in claudin-3 indicate that membrane-protein function imposed evolutionary restraints on native palmitoylation sites. These results suggest a generic, stochastic membrane-protein palmitoylation process that is determined by the accessibility of palmitoyl-acyl transferases to cysteines on membrane-embedded proteins, and not by a preferred substrate-sequence motif.

---

[1] Crystal and Structural Chemistry, Bijvoet Center for Biomolecular Research, Dept. of Chemistry, Faculty of Science, Utrecht University, Padualaan 8, 3584CH Utrecht, The Netherlands. [2] Biomolecular Mass Spectrometry and Proteomics, Bijvoet Center for Biomolecular Research and Utrecht Institute for Pharmaceutical Sciences, Faculty of Science, Utrecht University, Padualaan 8, 3584CH Utrecht, The Netherlands. Correspondence and requests for materials should be addressed to A.J.R.H. (email: a.j.r.heck@uu.nl) or to P.G. (email: p.gros@uu.nl)

Post-translational, covalent addition of palmitate to cysteine residues dynamically sorts cytosolic proteins and distributes membrane proteins within the membrane. Palmitoylation induces membrane attachment of the soluble proteins, whereas subsequent enzymatic depalmitoylation releases the proteins from the membrane[1, 2]. Palmitoylation can influence membrane-protein conformation, protein–protein interactions, and distribute membrane proteins over membrane domains[3, 4]. Changes in palmitoylation have functional implications on membrane proteins, such as altered receptor signaling[5, 6], reduced transporter activity[7], modified trafficking[8–10], and ion-channel regulation[11–13]. Palmitoylation is mediated by membrane-embedded palmitoyl-acyl transferases (PAT) that contain a palmitoylated cysteine-rich domain with a conserved Asp-His-His-Cys (DHHC) motif, which is required for the palmitoylation activity[14–18]. Disrupted protein palmitoylation by altered PAT activity has been implicated in various diseases, including cancers and neurodegenerative disorders[19].

The human genome encodes 23 DHHC-domain containing proteins[16]. A short amino-acid stretch is often sufficient to induce palmitoylation of cytosolic proteins[20, 21] and the motif inducing palmitoylation of Kv1.1-potassium channels is conserved in three other palmitoylated membrane-protein classes[22], suggesting PATs recognize sequence motifs present in substrates. However, incorporation of D-amino acids around the palmitoylated cysteine in the peripheral membrane-protein N-Ras did not prevent palmitoylation[21]. Even though some motifs have been linked to protein palmitoylation, there is no strictly conserved sequence motif[23–27]. For membrane proteins, palmitoylated cysteines are generally located at the cytoplasmic side or in transmembrane helices at the inner leaflet of the membrane[23–26, 28]. Membrane proteins often contain multiple cysteines at these locations that may be palmitoylated concomitantly[29]. Pulse-chase data on calnexin suggested a preferred order to produce dual-palmitoylated species[30]. Thus, to which extent PATs recognize substrates and the mechanism by which proteins are decorated with one or multiple palmitoyl chains still remains relatively poorly understood.

Protein palmitoylation remains generally elusive in shotgun proteomics experiments, since palmitoylated peptides do typically not make it to mass spectrometric detection, due to their hydrophobicity, general insolubility, poor fragmentation behavior and hydrolysis susceptibility of the thioester-linked palmitate[15, 31]. Moreover intramolecular palmitoyl transfer may occur from cysteine to lysine side chains or the protein N-terminus[32]. To overcome these limitations, chemical approaches have been developed to detect protein palmitoylation. Acyl-biotin exchange allowed for the identification of 50 palmitoylated proteins in yeast[18]. Metabolically adding 17-octadecynoic acid, a click chemistry probe for in situ labeling, allowed the identification and verification of hundreds of palmitoylated proteins in human cells[33, 34]. In addition, protein engineering[35] and replacement of palmitates by PEG have been employed to study protein palmitoylation[36, 37]. However, the lack of methods to detect palmitoylation of native proteins has hindered detailed studies on the stoichiometry of palmitoylation in membrane proteins with multiple palmitoylation sites and on membrane-protein palmitoylation specificity.

Here, we use native mass spectrometry (MS) to quantify the palmitoylation states of three members of the four-helical tight-junction protein family claudins (Cld), Cld3, Cld4, and Cld6, the four-helical B-lymphocyte antigen CD20 and cysteine-engineered constructs of the light-activated proton-pump bacteriorhodopsin and the potassium-channel KcsA. Earlier proteomics studies have identified claudins[38–41] and CD20[42–44] as palmitoylated proteins. Site-specific mutagenesis verified that rat Cld7 is palmitoylated on one to two cysteines in transmembrane helix 4[45] and Cld14 at least on two cysteines located in its intracellular loop and helix 4[46]. CD20 is palmitoylated at Cys 111 and/or Cys 220 located in the intracellular loop and in the cytoplasmic C-terminal tail, respectively[42]. In this study we analyzed palmitoylated membrane proteins using native MS. Detergent-solubilized membrane proteins were brought into the gas phase through nano-electrospray ionization[47]. The micelle protects the protein during transfer, next it is released from the protein using gas-phase activation, providing a means to determine the mass of the intact protein, even when it is modified by multiple palmitates. Using MS and site-directed mutagenesis we identify the palmitoylation sites in membrane proteins and determine the amount of palmitoylation at multiple co-occurring sites. Based on our data we propose a model in which palmitoylation of cysteines is not dependent on a local sequence motif. In these cases, modification of cysteines is rather determined by membrane-protein structure, in that it occurs generally on accessible residues within a depth of 8 Å of the lipid bilayer. Moreover, palmitoylation at multiple sites on Cld3 and CD20 is a stochastic but cooperative process. Collectively, these results provide insights into a membrane-protein palmitoylation process that is independent of a substrate-sequence motif.

## Results

**Mass spectra of micelle-released intact wild-type claudins.** Cld3, 4 and 6 were expressed and purified from human cells (HEK-293) with C-terminal His tag (Supplementary Fig. 1). A C-terminal fragment of *Clostridium perfringens* enterotoxin (C-CPE) was added to the culture medium to increase expression and prevent multimerization of claudins. C-CPE binds specific claudins with nanomolar affinity and removes them from tight junctions in vivo[48]. Native mass spectra of Cld3–C-CPE in micelles revealed peaks matching the theoretical masses of Cld3 with its initiator methionine removed and of C-CPE, as predicted based on amino-acid sequence (Fig. 1a, b; Supplementary Table 1). In addition to the theoretical masses matching the bare protein sequence, peaks for five other masses were observed for Cld3. Each consecutive peak corresponded to a relative mass increase of 238 Da, indicating Cld3 is modified with zero to five palmitoyl chains. The most prevalent Cld3 isoform is modified with four palmitates, suggesting that four dominant sites are available for palmitoylation. Cld3 purified without C-CPE addition yielded very similar mass spectra, indicating that C-CPE does not substantially alter palmitoylation of Cld3 (Supplementary Fig. 1b). Mass spectra obtained for Cld4 and Cld6 also showed similar distributions of variable palmitoylations, indicating that these claudins also contain multiple highly occupied palmitoylation sites (Fig. 1c).

Cld3 contains five putative palmitoylated cysteines in transmembrane helices at the height of the inner-membrane leaflet and in the intracellular loop (Cys 103, 106, 181, 182, and 184). Mutation of all putative palmitoylated cysteines to alanines (hereafter referred to as Cld3-ΔCys) abolished the majority of Cld3 palmitoylation; however, ~10% of Cld3 still contained a palmitate moiety (Supplementary Fig. 2). This low-abundant palmitoylation isoform was not observed when Cld3 was produced with native termini, indicating that the additional palmitoylation was a consequence of our cloning strategy. Nevertheless, we used the His-tagged claudin constructs throughout the study, because of higher purification yields. Next, the putative palmitoylated cysteines were individually mutated to alanine to identify the four dominant palmitoylations sites. Mutation of Cys 181 did not change the palmitoylation distribution observed for native Cld3, whereas mutations of Cys 103, 106, 182, or 184 shifted the distribution in a manner

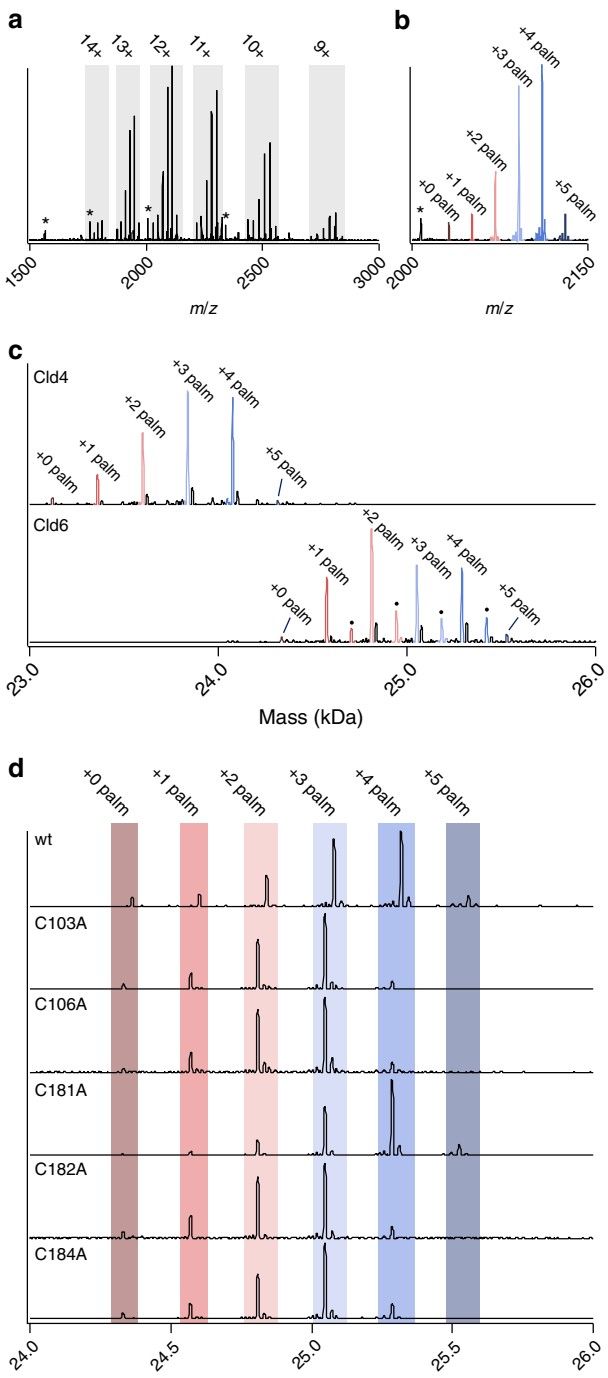

**Fig. 1** Native mass spectra of micelle-released poly-palmitoylated claudins. **a** Spectrum of Cld3 in complex with C-CPE, showing a mass distribution corresponding to Cld3 harboring zero to five attached palmitates. Gray boxes contain Cld3 peaks with the corresponding charge state annotated. Peaks labeled with an asterisk represent C-CPE peaks. All panels in this figure show representative data from biological triplicates. **b** Close up of the 12 + Cld3 charge state with the number of palmitates bound to Cld3 annotated for each peak. **c** Deconvoluted native ESI-MS spectra of poly-palmitoylated Cld4 and Cld6. The Cld6 peaks annotated with a dot correspond to a Cld6 mass without its initiator methionine removed. **d** Deconvoluted spectra of native Cld3 and of Cld3 with putative palmitoylated cysteines mutated to alanine one by one, showing that mutation of Cys 103, 106, 182, or 184 to Ala result in the loss of a single palmitate, whereas mutation of Cys 181 does not

consistent with the elimination of a single-palmitoylation site, identifying these cysteines as the dominant native palmitoylated residues (Fig. 1d). Cysteines are conserved at equivalent positions in Cld4 and Cld6, which explain the observed, similar distribution of palmitoylation of these related claudins (Supplementary Fig. 4).

We tested the effect of protein expression levels on the observed palmitoylation and the palmitoyl-thioester stability upon storage. Three-fold to five-fold reduced expression levels of wild-type Cld3, Cld4, and Cld6 did not change palmitoylation levels (Supplementary Fig. 3a). Re-analysis of purified Cld3, Cld4, and Cld6 after storage for one week in detergent buffer used for MS did not substantially change the palmitoylation distribution (Supplementary Fig. 3b), indicating that attached palmitates are stable in the micelle environment.

**Cysteine scanning reveals putative palmitoylation sites**. To gain insight into the specificity of membrane-protein palmitoylation, we introduced single Cys mutations in Cld3-ΔCys at 27 different positions. Cysteines were incorporated at sites located in the cytoplasmic region and at the inner-membrane leaflet of the four transmembrane helices and the intracellular loop. These Cld3 mutants were expressed and purified in complex with C-CPE from HEK cells and their palmitoylation levels were determined by native MS. Two mutants, Ser 2 Cys and Gly 100 Cys, were excluded from this analysis due to dimer formation and lack of C-CPE binding, respectively. Palmitoylation was observed at 18 sites (Fig. 2a), of which 14 are novel palmitoylation sites and four sites are palmitoylated in wild-type Cld3 as well (Fig. 1d). The 14 novel palmitoylation sites, introduced by Cys mutations, are positioned on all four transmembrane helices of Cld3.

Sequence alignment of Cld3 with other classic claudins[49] shows that the four native palmitoylation sites of Cld3 are strongly conserved (Supplementary Fig. 4). Several novel sites that we found by cysteine scanning correlate with non-conserved cysteines in classic claudins, i.e., Cld8 (res. 99, 185, and 186; using Cld3 numbering throughout), Cld15 (res. 185 and 186) and Cld17 (res. 185 and 186). Consistent with the known location of palmitoylation sites in Cld7[45] and Cld14[46], all sites in classic claudins cluster in the intracellular loop and helix 4. Most of the putative palmitoylated cysteines in non-classical claudins also map to these regions (Supplementary Fig. 5). However, non-classic Cld11 and Cld18 contain a cysteine in helix 1 at residues 4 and 5, respectively. At these positions, cysteines were palmitoylated in Cld3-cysteine scanning. Thus, except for 2 non-classic claudins, the palmitoylated sites cluster in the intracellular loop and in helix 4 and no cysteines occur in other helices that can be palmitoylated.

**Cysteines up to 8 Å into the membrane can be palmitoylated**. Palmitoylation levels of cysteine-scanned residues were mapped on a Cld3 homology model generated based on the crystal structure of mCld15[50] (Fig. 2b, Supplementary Fig. 6). This lipid-based crystallization of mCld15 is currently the claudin structure with the highest resolution available, and exhibits a physiologically relevant tight-junction arrangement. The depth of each palmitoylation site into the membrane (defined as the approximate Cβ to inner-leaflet phosphor distance) was determined using the Cld3 model in membrane orientations reported for mCld15 and mCld19[51]. The mCld15 orientation was derived from its lipid-based crystal structure[50] and the orientation of mCld19[51] was obtained from simulation in a lipid bilayer by memprotMD[52]. In both claudin orientations all palmitoylated residues are located within 8-Å depth into the inner leaflet of the membrane (Fig. 2c; Supplementary Fig. 7a). Four cysteines introduced in positions located further than 8 Å into the membrane in both claudin orientations were not palmitoylated.

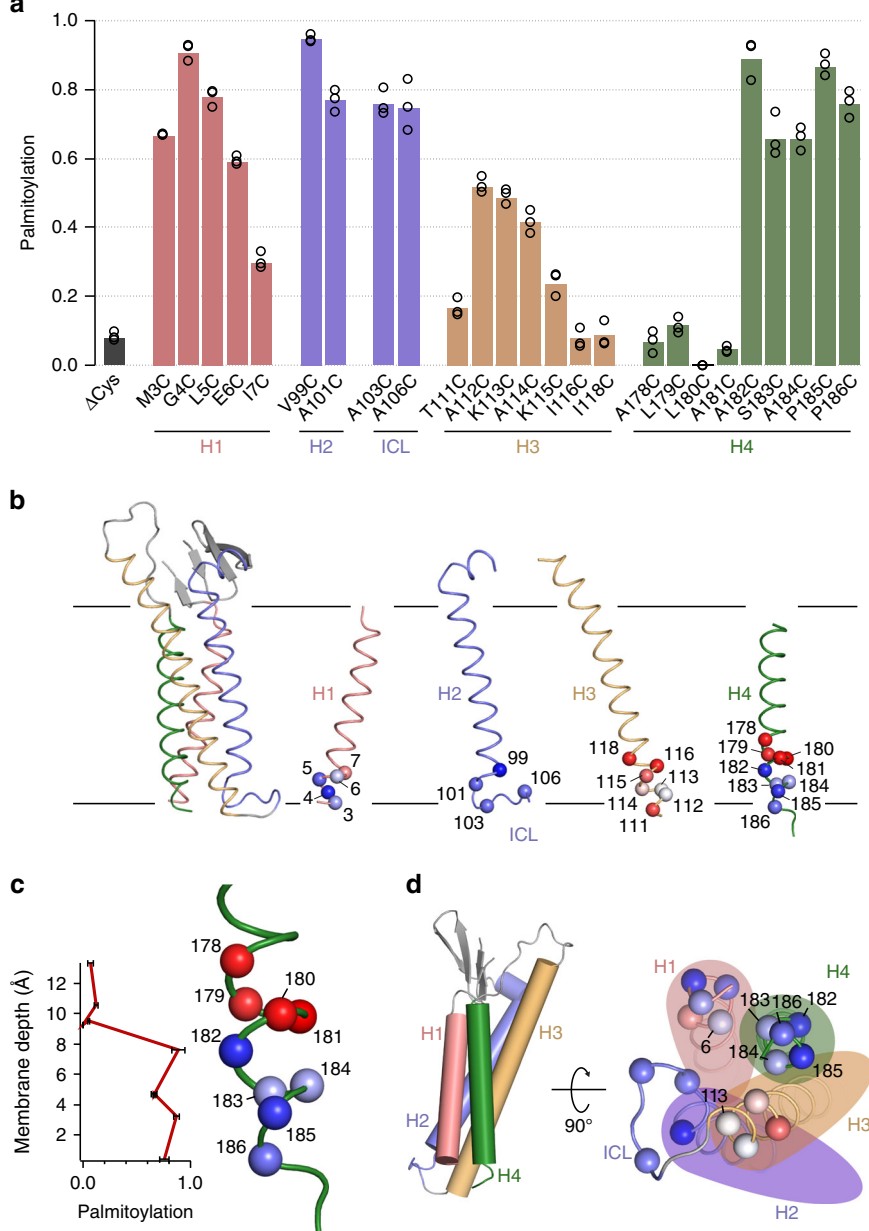

**Fig. 2** Palmitoylation status of cysteine-scanned Cld3. **a** Palmitoylation of single cysteines introduced in Cld3 with all putative palmitoylated Cys 103, 106, 181, 182, and 184 mutated to Ala (ΔCys). The average palmitoylation state is obtained by using integrated peak areas of the deconvoluted spectra of intact micelle-released Cld3. No palmitoylation could be observed for Leu 180 Cys because of low expression of this mutant. A palmitoylation state of 0.2 was used as threshold value to classify the introduced Cys as palmitoylation site. Data are presented as mean values of biological triplicates and independent data points are shown as open circles. **b** Introduced cysteine residues mapped on the Cld3 homology model. In all panels, the Cα atom of each cysteine is represented as a sphere colored in a red–white–blue gradient normalized according to its palmitoylation level, in which full palmitoylation is shown as blue, 50% palmitoylation as white no palmitoylation as red. **c** Cysteine residues introduced in helix 4 are not palmitoylated when located further than 8 Å into the membrane. The membrane depth is the measured distance of the Cα atom of the introduced cysteine to the average position of all phosphor atoms of the inner leaflet after superimposition of the Cld3 model to mCLD19 in MemprotMD. Palmitoylation data are presented as mean values of biological triplicates and error bars represent standard deviations. **d** Left panel, structural side view of Cld3. Right panel, cytoplasmic view of Cld3 with palmitoylated Cα atoms in shown as spheres. The majority of palmitoylated residues are exposed

The majority of palmitoylated residues are exposed to the membrane or the cytoplasm in the Cld3 homology model (Fig. 2d). However, palmitoylation was also observed on buried residue 6 in helix 1, residue 113 in helix 3, and residues 183 and 184 in helix 4. Loss of charge upon mutation of Glu 6 and conserved Lys 113 might destabilize local structure such that this buried site becomes available for palmitoylation. Mutation of Ser 183 and Ala 184 to cysteine resulted in less palmitoylation

compared to surface exposed neighboring residues 182 and 185, suggesting that exposed residues are more likely to be palmitoylated (Fig. 2a, Fig. 3b).

Overall, cysteines introduced in helix 3 were markedly less palmitoylated compared to cysteines in other helices (Fig. 2b), despite the fact that these are located within 8-Å depth into the membrane. Three introduced cysteines that are not palmitoylated are located in this helix. The membrane orientation of mCld15[50]

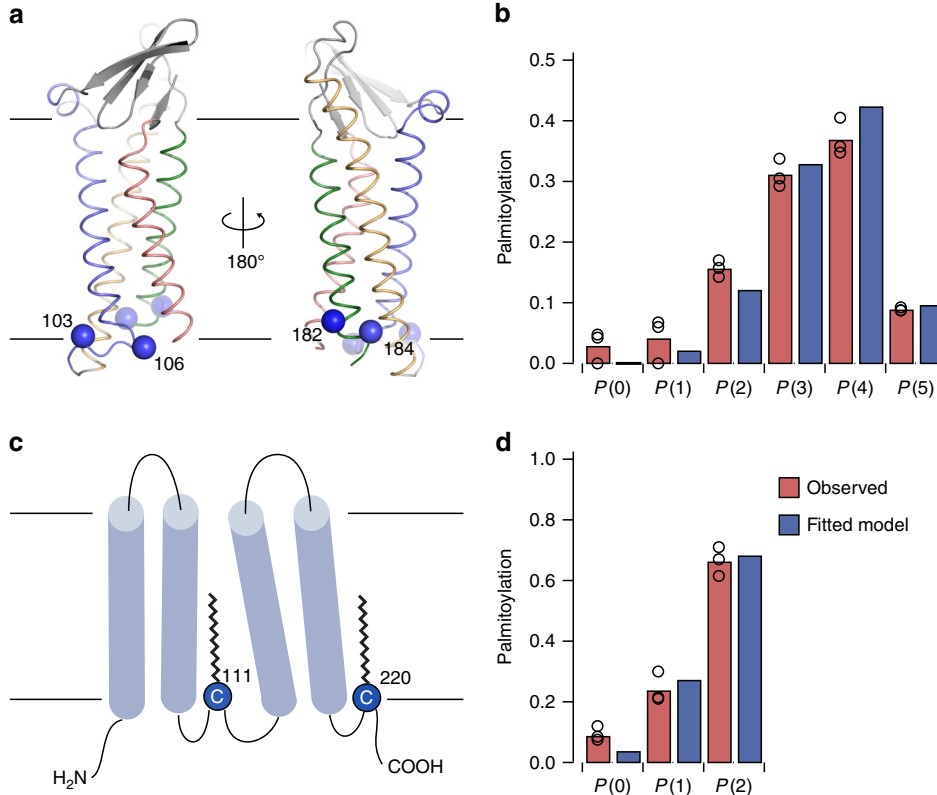

**Fig. 3** Quantitative analysis of Cld3 and CD20 palmitoylation. Cartoon representation of the Cld3 model (**a**) and schematic representation of CD20 (**b**). Probability of cysteines to be palmitoylated in Cld3 is $p_{103} = 0.69$, $p_{106} = 0.69$, $p_{182} = 0.86$, $p_{184} = 0.64$ and in CD20 is $p_{111} = 0.74$, $p_{220} = 0.82$. Cβ atoms of palmitoylated cysteines shown as spheres and colored according to the probability to be palmitoylated in a white–blue gradient, in which full palmitoylation is shown as blue and no palmitoylation as white. **b**, **d** Comparison of the observed and calculated protein fractions for each palmitoylation state (P($n$)) of wild-type Cld3 (**b**) and CD20 (**d**) after fitting the cooperative stochastic model to palmitoyl distributions of all Cld3 or CD20 constructs

suggests claudins adopt a tilted orientation in the membrane. In this orientation, transmembrane helices 1, 2, and 3 are tilted by 24°, 32°, and 23°, respectively, whereas helix 4 is oriented almost parallel to the membrane normal (3° tilt). For helix 3, the palmitoylation sites lie on a lipid-exposed surface, which forms an acute angle with the bilayer plane (Supplementary Fig. 7b). These data suggest that the claudin tilt obstructs exposed residues on helix 3 to become palmitoylated.

Cysteine palmitoylation does not occur in prokaryotes and DHHC-domain containing proteins are not found in prokaryotic genomic databases[16]. We attempted to palmitoylate prokaryotic membrane proteins in human cells, to be able to rule out that palmitoylation requires other signals, beside an available cysteine. Palmitoylation sites were introduced in two cysteine-free prokaryotic membrane proteins: the light-activated proton-pump bacteriorhodopsin and potassium-channel KcsA. We incorporated cysteines specifically at lipid-exposed positions located up to 4 Å into the membrane, based on the crystal structures of monomeric bacteriorhodopsin[53] and tetrameric KcsA[54] placed in membranes[52]. Retinal-free bacteriorhodopsin and N-terminally truncated KcsA (to increase purification yields), were expressed in human cells (Supplementary Fig. 8). Upon addition of retinal during purification, bacteriorhodopsin mass spectra indicated peaks corresponding to the mass of a retinal-bound form and KcsA migrated as a tetramer on SDS-PAGE gel (Supplementary Fig. 1), indicating that both proteins and mutants were expressed in their functional form. Bacteriorhodopsin palmitoylation was observed on 5 out of 6 introduced cysteines, each located at a lipid-exposed position on one of its transmembrane helices based on a retinal-bound bacteriorhodopsin structure[53] (Supplementary

Fig. 8c). In the retinal-free state of bacteriorhodopsin, the state in which palmitoylation occurs, Phe 153 Cys might be in a less exposed conformation precluding palmitoylation. Both cysteines introduced in or near the lipid-exposed helix 1 of KcsA were found to be palmitoylated (Supplementary Fig. 8d). Since no cysteine palmitoylation occurs in prokaryotes[16], the ability to palmitoylate these prokaryotic proteins strongly suggests that a single exposed cysteine at the right depth in the membrane is the sole requirement to induce palmitoylation.

**Cld3 and CD20 palmitoylation is stochastic and cooperative.** We quantitatively analyzed the distributions of multiple palmitates observed for wild-type Cld3. Besides wild-type Cld3, we included all possible combinations of Cld3 mutants with either three or a single native palmitoylation site into the analysis, generating multiplicity in the data for accurate determination of model parameters. As described above, the proteins exhibited an additional low-abundance palmitoylation site due to our cloning strategy. First, we considered palmitoylation reactions as independent events, i.e., each site having an individual palmitoylation probability accounting for effects of local structure. These 'single-site' probabilities were obtained from the observed palmitoylation rates of single-cysteine mutants (Cys 103, 106, 182, and 184) and a variant without cysteines in the intracellular loop and helix 4 (Cld3-ΔCys), accounting for the low-abundance palmitoylation site. We used the observed single-site palmitoylation frequencies as prior probabilities and computed the predicted distribution of multiple palmitates on Cld3 (Methods section). The overall correlation between the predicted and the observed distributions was $R^2 = 0.91$ for 18 independent experiments with 96 observations in

total (Supplementary Fig. 9a, b). Second, incorporation of one additional cooperativity parameter $c$, which models the increased chance for subsequent palmitoylation events, improved the prediction to $R^2 = 0.96$. Fitting all six parameters (five single-site probabilities and the cooperativity parameter) of the cooperative model to the Cld3 data yielded $R^2 = 0.99$ with a cooperativity parameter of 1.24 and minor modifications (up to 0.04) of the single-site probabilities (Fig. 3a, b; Supplementary Fig. 10a, Supplementary Table 2).

Next, we predicted the palmitate distribution of doubly palmitoylated CD20 using probabilities derived from mutants containing single-palmitoylation sites (Cys 111 Ala and Cys 220 Ala; Supplementary Fig. 11). A correlation of $R^2 = 0.96$ was obtained when treating palmitoylation reactions as independent events (Supplementary Fig. 9c, d). With two palmitoylation sites per protein molecule only, a single experiment (i.e., that of doubly palmitoylated native protein) yields information on the cooperativity. Assuming a cooperativity of 1.24 as determined for Cld3, improved the prediction of the native-CD20 palmitate distribution ($R^2 = 0.98$), suggesting CD20 palmitoylation is also a cooperative process. Fitting the cooperative model to all CD20 data ($R^2 = 0.99$; Fig. 3c, d, Supplementary Fig. 10b) resulted in a cooperativity parameter of 1.51 (Supplementary Table 2). In summary, these data indicate that palmitoylation of Cld3 and CD20 appears to be governed by a stochastic and weakly cooperative process.

## Discussion

We demonstrated that transferring micelle-solubilized and purified membrane proteins into the gas phase enables the efficient detection of protein palmitoylation. Native MS of micelle-released intact-palmitoylated membrane proteins does not require the use of labels or chemical reactions during sample preparation. The micelles protect the labile thioester, between the palmitate moiety and the protein from hydrolysis. Thus, native MS of detergent-solubilized membrane proteins uniquely allows for a quantitative determination of the palmitoylation stoichiometry. This overcomes problems encountered in standard peptide-centric proteomics workflows, due to poor recovery of hydrophobic palmitoylated-tryptic peptides. However, the presented method requires purified membrane proteins stabilized in micelles suitable for gas-phase ionization. When these technical requirements are fulfilled[55], native MS provides an approach to investigate a wide range of post-translational modifications of this class of proteins, such as putative attachments of acyl chains of different lengths or the interplay of palmitoylations as studied here.

Our native MS analysis of cysteine palmitoylation revealed that cysteines on Cld3 situated maximally 8 Å into the inner-membrane leaflet and exposed to the lipid environment become palmitoylated. This distance corresponds to 5–6 residues for a helix perpendicular in the membrane. Native claudins are palmitoylated almost exclusively in the intracellular loop and helix 4, however cysteine scanning of Cld3 identified palmitoylation on all four transmembrane helices. Lower palmitoylation levels were observed for cysteines introduced in helix 3. At this helix the protein forms an acute angle with the membrane, putatively limiting lateral access of membrane-embedded PATs and thereby reducing palmitoylation locally (Fig. 4). Moreover, palmitoylation of cysteines introduced in prokaryotic membrane proteins bacteriorhodopsin and KcsA expressed in HEK cells demonstrates that palmitoylation of these proteins does not depend on recognition of a specific sequence or structural motif. These data are in line with observations on single-pass type-II membrane proteins that the specific local amino-acid sequence is not essential for palmitoylation[23–26]. In vivo palmitoylation is

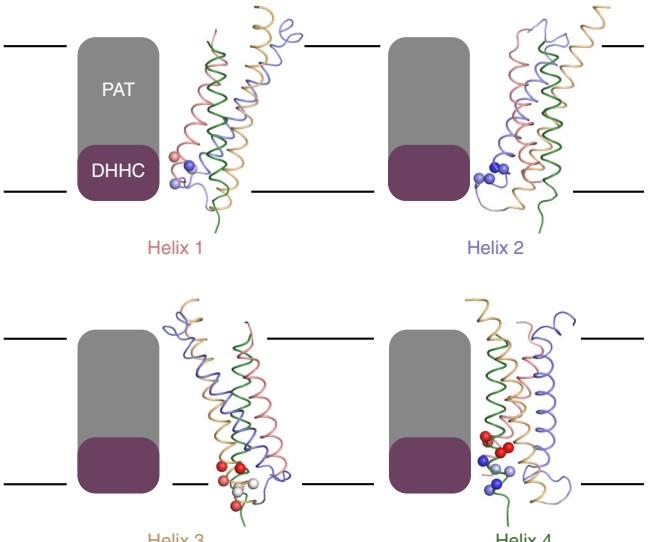

**Fig. 4** Cysteine palmitoylation is restricted by palmitoyl-acyl transferases (PAT) accessibility. PATs are restricted to lateral diffusion for its catalytic DHHC domain to interact with substrates. The membrane orientation of Cld3 prevents PATs to access palmitoylation sites located on helix 3, whereas palmitoylation sites on helix 1, 2, and 4 are accessible

expected to be enzymatic[16–18]. Although our data do not exclude a non-enzymatic process, the implications remain identical, i.e., palmitoylation of free, accessible cysteines may occur up to 8 Å into the membrane on the cytosolic side and palmitoyation may occur independent of a sequence motif, but is governed by the accessibility of the free cysteine.

Membrane proteins carrying multiple palmitoylation sites give rise to a distribution of protein isoforms. The distributions for Cld3 and CD20 contain isoforms with none to all sites palmitoylated, i.e., four and two native palmitoylation sites, respectively. Analysis of these proteins with multiple palmitoylation sites and their single-site mutants showed that the observed distributions of palmitoylation isoforms can be predicted with high accuracy from the single-site palmitoylation rates (96 and 98% for Cld3 and CD20, respectively). The accuracy of this prediction implies that there are no potential differences in subcellular localization of mutants vs. wild-type Cld3 and CD20 that affect the overall accuracy of the palmitoylation measurement. Single-site rates are likely determined by the PAT-accessibility of a particular cysteine. Moreover, the quantitative analysis demonstrates that membrane-protein palmitoylation at multiple sites in Cld3 and CD20 is a stochastic process (i.e., consisting of a series of independent events). The process yielding multiple palmitoylations is, however, weakly cooperative, i.e., the chance of subsequent palmitoylation increases by ca. 25%. Such a weak cooperativity may be explained by locally increased levels of PAT enzymes present due to prior palmitoylation of the multiply palmitoylated membrane proteins. Depalmitoylation of membrane proteins has been observed for sites on intracellular loops[56] and flexible termini[57], not for sites in and near transmembrane helices such as those studied here. Although we cannot exclude depalmitoylation occurring, our data is consistent with stochastic palmitoylation both with and without potential stochastic depalmitoylation. However, our observations of stochastic palmitoylations are in stark contrast to sequential palmitoylation reported for two neighboring cysteines in calnexin[30]. Possibly, specific palmitoylation for calnexin by DHHC6[58] and non-specific stochastic palmitoylations, as described here, are distinct cellular processes.

Most protein post-translational modification are highly sequence specific, e.g., N-linked glycosylation[59] and phosphorylation[60], preventing modification at unwanted positions. In contrast, here we show that even prokaryotic membrane proteins can be palmitoylated in human cells simply by introducing cysteines. In classic claudins, palmitoylated cysteines occur only in the intracellular loop and helix 4 at the flanks of tight junctions (Supplementary Fig. 12); however, cysteine scanning revealed that claudins can be palmitoylated at all transmembrane helices. Palmitoylation at other helices possibly interferes with claudin function, e.g., palmitates attached to helix 1 possibly obstruct claudin-claudin interactions needed for tight-junction formation. Therefore, evolutionary restraints likely prevented cysteine incorporation at undesired positions to avoid deleterious palmitoylation. Acyl chains of different lengths may be attached to cysteines in membrane proteins[61]. However, we detected palmitate modifications only. What triggers potentially differential lipidation of membrane proteins, remains to be determined. Moreover, our data does not exclude the existence of other, potentially sequence specific, lipidation, and delipidation mechanisms with possible regulatory functions.

In conclusion, based on our study we propose a model for generic, non-specific palmitoylation of membrane proteins. In contrast to most other protein post-translational modifications, our model suggests that generic membrane-protein palmitoylation is not determined by a specific sequence motif, but merely by the accessibility of the substrate cysteine to the catalytic center of the PAT enzymes. Non-specific palmitoylation of multiple sites on membrane proteins does not follow a predefined sequential order, but is rather determined by a stochastic process yielding a distribution of palmitoylated isoforms. Even though this process is stochastic, palmitoylation of multiple sites on a membrane protein also exhibits a weak cooperativity. The rate of subsequent palmitoylations is slightly increased, because prior palmitoylation increases the likelihood of PAT enzymes in the vicinity of the substrate protein. The process of palmitoylation and depalmitoylation is both dynamic and highly complex and therefore requires both system wide and protein specific studies. The model proposed here provides detailed insight in the palmitoylation specificity of a set of proteins and suggests the existence of a generic, non-specific mechanism for palmitoylation, besides possibly other specific mechanisms. Future experiments testing the applicability of this model on proteins in their native environment are required to unravel the workings of dynamic palmitoylation and depalmitoylation.

## Methods

**Protein expression and purification**. Codon optimized DNA for bacterial expression of the C-terminal fragment (residues 197-319) of CPE was obtained from GeneArt and cloned into a pETx expression vector containing a N-terminal His$_6$ followed by a tobacco etch virus protease (TEV) recognition site and expressed in BL21(DE3) E. coli cells. Cell pellets were resuspended in 25 mM HEPES pH 7.5, 200 mM NaCl, 10 mM MgCl$_2$, and lysed by repeated freeze-thaw cycles with addition of lysozyme (Sigma Aldrich) and DNAseI (Sigma Aldrich). Cell debris was removed by centrifugation (9500 × g, 1 h), imidazole was added to a final concentration of 20 mM and Ni-sepharose beads (GE Healthcare) were added. Beads were washed with 25 mM imidazole and eluted with 250 mM imidazole. 0.5 mg ml$^{-1}$ His$_6$ tagged TEV was added to eluted protein 1:10 (v/v), after which the sample was dialyzed against 25 mM Tris-HCL pH 8.0, 200 mM NaCl and 5% glycerol. TEV was removed by passing the protein over a 5 ml HisTrap HP column (GE Healthcare). Purified C-CPE was filter sterilized and flash frozen until use.

Codon optimized DNA without start and stop codon of claudins and CD20 was obtained from DNA 2.0 and Bacteriorhodopsin (Halobacterium salinarum) DNA from GeneArt. KcsA (Streptomyces lividans) was amplified from residue 22–160 without stop codon and inserted into pCR8 vector using TOPO TA cloning (Thermo Fischer). For mammalian cell membrane-protein expression, DNA was sub-cloned in an expression vector containing a start codon and a C-terminal His$_6$ tag (claudins) or a C-terminal triple repeat StrepII tag (bacteriorhodopsin, KcsA and CD20) using 5′ BamHI and 3′ NotI restriction sites. Proteins were transiently expressed in HEK293-EBNA1 suspension cells supplied by U-Protein Express BV,

authenticated by STR profiling (BaseClear) and tested for mycoplasma contamination (BaseClear). C-CPE (0.45 μM) was added to the culture medium of cells expressing claudins (no C-CPE was added to obtain Cld3 without C-CPE). Protein yields were optimized by plasmid titration with a dummy plasmid expressing the tripeptide MGS[62]. Dilution of the protein encoding plasmid 100-fold (w/w) generally resulted in highest membrane-protein expression. For reduced levels of claudin expression, claudin plasmids were diluted 500-fold and 1000-fold with dummy plasmid.

For claudin purification, 96 h after transfection, cell pellets of cultures were solubilized in 25 mM HEPES pH 7.5, 25 mM NaCl, 1% (w/v) n-dodecyl-β-D-maltopyranoside (DDM, Anatrace), EDTA-free protease inhibitor cocktail (Roche) and 2.3 μM C-CPE for 1 h at 4 °C. After removal of insoluble material by ultracentrifugation (100,000 × g, 1 h, 4 °C), imidazole was added to a final concentration of 25 mM, the detergent concentration was adjusted to 0.3% (w/v) and Ni Sepharose HP beads (GE healthcare) were added (no C-CPE was added to obtain Cld3 without C-CPE). Beads were washed with 25 mM HEPES pH 7.5, 25 mM NaCl, 0.03% (w/v) DDM, 35 mM imidazole, and purified claudins were eluted with 375 mM imidazole in 60 ul. CD20, Bacteriorhodopsin and KcsA were purified according to the same protocol with the following modifications. No C-CPE was added in the culture medium and during solubilization and no imidazole was added throughout the purifications. After detergent dilution, StrepTactin Sepharose HP beads (GE healthcare) were added and protein was eluted with 2.5 mM D-Desthiobiotin (Sigma).

To obtain Cld3 with native termini, a Cld3 construct with native start and stop codon was cloned in an expression vector lacking start and stop codons. Expression and purification were done as described above for claudins with the following modifications, instead of tagless C-CPE, 0.45 μM unprocessed C-CPE (His-TEV-C-CPE) was added to the culture medium, prior to solubilization cells were washed with phosphate buffered saline and no C-CPE was added during solubilization.

**Mass spectrometry**. Purified membrane proteins were buffer exchanged to 150 mM ammonium acetate pH 7.5, 0.025% (w/v) DDM with Micro Bio-Spin columns (Biorad). An aliquot of 1–2 μl were loaded into gold-coated borosilicate capillaries (prepared in-house) for nano-electrospray ionization. Samples were analyzed on a modified Orbitrap EMR instrument (Thermo Fisher Scientific)[63, 64]. Capillary voltage was set at 1200 V, source fragmentation 50 V, extended trapping 200 V. Nitrogen was used as collision gas. Spectra were acquired using an acquisition time of 32 or 64 ms per scan, averaged over several scans, typically one minute. Mass assignments were performed manually using XCalibur software (Thermo Fisher Scientific), relative levels of palmitoylation were extracted from deconvoluted spectra using Protein Deconvolution 3.0 (Thermo Fisher Scientific) or Masslynx 4.0 Transform (Waters).

**Mutagenesis**. All mutants were generated with Phusion Site-Directed Mutagenesis Kit (Thermo Fisher).

**Homology modeling**. The Cld3 sequence was aligned to the crystallized sequence of mCld15[50] (PDB code 4p79) using ClustalW[65]. After manual correction of the alignments, 5 homology models were built with MODELLER v 9.13[66] using the mCld15 structure as template. After removal of poorly modeled regions (residues 33–43 and residues from 188 onwards), due to lack of electron density in the crystal structure of the mCld15 template, a single representative model was used for analysis.

**Quantitative analysis**. We developed non-cooperative and cooperative stochastic model to explain the observed protein fractions for each palmitoylation state of Cld3 constructs containing any, but non-other than the palmitoylation sites present in native Cld3. The observed protein fraction of palmitoylation state n is described by the probability for Cld3 to obtain n palmitates, represented as $P(n)$ for the non-cooperative and $P^{coop}(n)$ for the cooperative model. In the independent model we assume that palmitoylation of different sites are assumed to be independent events. Thus, $P(n)$ is a function of the palmitoylation probabilities $p_i$ of the palmitoylation sites i present in the modeled Cld3 construct and is given by the sum of the probabilities of all possibilities to obtain n palmitoylations, as illustrated for a Cld3 construct with three palmitoylation sites below.

$$\overline{p_i} \equiv 1 - p_i$$

$$P(0) = \overline{p_1} \cdot \overline{p_2} \cdot \overline{p_3}$$

$$P(1) = p_1 \cdot \overline{p_2} \cdot \overline{p_3} + \overline{p_1} \cdot p_2 \cdot \overline{p_3} + \overline{p_1} \cdot \overline{p_2} \cdot p_3$$

$$P(2) = p_1 \cdot p_2 \cdot \overline{p_3} + p_1 \cdot \overline{p_2} \cdot p_3 + \overline{p_1} \cdot p_2 \cdot p_3$$

$$P(3) = p_1 \cdot p_2 \cdot p_3$$

To incorporate cooperativity we introduced the conditional probability to obtain n+1 palmitoylations if n palmitoylations are present, depicted as $P(n+1|n)$

for the independent model, as

$$P(n+1|n) \equiv \frac{P(n+1)}{P(n)},$$

and similarly for the cooperative model,

$$P^{\mathrm{coop}}(n+1|n) \equiv \frac{P^{\mathrm{coop}}(n+1)}{P^{\mathrm{coop}}(n)}.$$

Next we defined cooperativity as

$$P^{\mathrm{coop}}(n+1|n) = c^n \cdot P(n+1|n)$$

Combining the above three equations yields

$$P^{\mathrm{coop}}(n) = N \cdot c^{\left(\sum_{i=0}^{n} i\right) - n} \cdot P(n).$$

In which $N$ is the normalization constant which ensures $\sum_{n=0}^{n_{\max}} P^{\mathrm{coop}}(n) = 1$, i.e.

$$N = \frac{1}{\sum_{n=0}^{n_{\max}} \left( c^{\left(\sum_{i=0}^{n} i\right) - n} \cdot P(n) \right)},$$

in which $n_{\max}$ is the maximum number of available palmitoylation sites in the modeled Cld3 construct.

We fitted the protein fractions, as calculated by independent (non-cooperative) and the cooperative stochastic model, to the observed protein fractions of single-cysteine Cld3 constructs with a cysteine at a native site and Cld3-ΔCys. To do so we applied least squares minimization of the differences between the calculated and observed fractions by optimizing the palmitoylation probabilities of the available palmitoylation sites $p_i$ in the measured construct and the cooperativity parameter $c$ if applicable using Mathematica (Wolfram Research). None of the parameters were restrained during the minimization. Next we used the parameters obtained by fitting the non-cooperative and the cooperative model to predict the protein fractions of Cld3 constructs containing multiple palmitoylation sites using the respective model. Finally, the cooperative model was chosen for optimization of all parameters against all claudin data to obtain the most accurate probability for each site to be palmitoylated.

The palmitoylation probabilities of CD20 were obtained by fitting the independent model to CD20 variants with a single cysteine available for palmitoylation. These probabilities were used to compute the protein fractions of native-CD20 containing 2 palmitoylation sites using the independent model. The cooperativity parameter cannot be derived from fitting the cooperative model to CD20 with a single available palmitoylation site. Therefore, we used the cooperativity parameter obtained from fitting the cooperative model to all claudin constructs to predict the CD20 wt spectrum. Next, the most accurate probability for each site to be palmitoylated was obtained by optimizing all parameters of the cooperative model to all CD20 data.

**Code availability**. The Mathematica code used to generate the reported results is available from the corresponding author upon request.

**Data availability**. The raw MS data that support the findings are available from the corresponding author upon request.

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

## Acknowledgements

We thank L.M.J. Kroon-Batenburg and A.J. Rodenburg (Utrecht University) for advice on cooperativity modeling and E.G. Huizinga (Utrecht University) for critically reading the manuscript. This research was financially supported by Focus & Massa: Life Sciences and Biocomplexity grant from the Utrecht University and by the Council for Chemical Sciences of the Netherlands Organization for Scientific Research (NWO-CW) grant Targeting Membrane Proteins (project 731.015.201). The mass spectrometry work was supported by Proteins At Work (project 184.032.201), a program of the Netherlands Proteomics Centre financed by the NWO as part of the National Roadmap Large-scale Research Facilities of the Netherlands and a Projectruimte grant (12PR3303-2) from Fundamenteel Onderzoek der Materie (FOM). P.G. and A.J.R.H. acknowledge support by the Institute for Chemical Immunology, an NWO Gravitation project funded by the Ministry of Education, Culture and Science of the Netherlands.

## Author contributions

R.N.P.R., J.S. and P.G. initiated the project. R.N.P.R. and A.S. developed membrane-protein purification protocols. R.N.P.R. purified all membrane proteins. R.N.P.R and J.G. performed site-directed mutagenesis. R.N.P.R. performed homology modeling. J.S. and M.v.d.W. collected and analyzed mass spectrometry data. R.N.P.R and P.G. fitted measured data to stochastic models. P.G. and A.J.R.H. provided overall project guidance. All authors discussed the results. R.N.P.R., J.S., A.J.R.H. and P.G. wrote the manuscript.

## Additional information

**Competing interests:** The authors declare no competing financial interests.

