## [Peer Review File · Nature Communications]

Reviewers' comments:

Reviewer #1 (Remarks to the Author):

Using a novel MS technique combined with mathematical modelling and mutagenesis analyses the authors address a number of issues regarding palmitoylation of integral membrane proteins, particularly in cysteines that are close to the cytosolic border of transmembrane domains. Overall, the work, which is technically challenging, has been performed meticulously and the conclusions are supported by the data.

An important problem encountered throughout the article is lack of reference and discussion of previous work in the field.

-Up to now, techniques to investigate the stoichiometry of s-acylation were not readily available and, in this respect, the paper represents a breakthrough. Nevertheless, stoichiometry of S-acylation has been achieved using alternative MS methods (Liang et al., 2002). This work should be cited and the improvements over these methods should be stated.

- The authors prove that cysteines that are deeper than 8Å within the membrane, are not palmitoylated, and this is of interest for the field. It would be useful if the authors could make a rough estimate, or discuss the 8Å distance correspondence, to an approximate position of a cysteine in a TMD, for membrane proteins in general. It is clear that the distance would depend on the tilt of a given TMD. Nevertheless, Assuming little to no tilt, up to which positions in the TMD can cysteines be palmitoylated?. This information, however approximate, would be very useful to the reader and it would allow comparison with existing data. Experimentally, it has been shown that cysteines in position 6 from the border toward the membrane are palmitoylated but cysteines that are located further are not (Ponimaskin and Schmidt, 1995).

-The authors demonstrate conclusively that it is the position of the cysteines rather than any sequence requirement that is necessary for cysteine palmitoylation. The use of a mutated prokaryotic protein to study the sequence requirements for protein palmitoylation is particularly clever. However, the work reinforces by novel and elegant techniques, ideas that have already been proposed, most importantly by (ten Brinke et al., 2002; Yik and Weigel, 2002) (Citations to these two last articles are misplaced in the text). These articles should be properly cited and discussed. Other authors have also address this issues, perhaps less directly, particularly in the field of viral protein palmitoylation, and more recently by (Gonzalez Montoro et al., 2011)

-(Dallavilla et al., 2016) have addressed the issue of cooperativity in the palmitoylation of cysteines that are close to each other and to the TMD for Calnexin, and they propose a high degree of cooperativity. This should be cited and discussed. It was difficult to discern whether the authors find differences in the cooperativity of palmitoylation between cysteines that are close to each other in a TMD or that are in different TMDs. This could be an interesting point to address.

-Palmitoylation is known to interfere with the subcellular localization of multispanning membrane proteins i.e. (Lam et al., 2006) reviewed in (Blaskovic et al., 2013). It is important that the subcellular localization of the mutant proteins is assessed. It is possible that mutants used in this study would be incorrectly localised and therefore inaccessible, or less accessible to its PAT. This would interfere with the interpretation of the proposed palmitoylation rates.

- An important aspect that has not been included in the arguments is de-palmitoylation. The profiles of palmitoylated proteins observed could very well be the result of a balance between acylation –deacylation. There are not many examples of integral membrane proteins that are actively de-palmitoylated (in cysteines close to their TMDs). Nevertheless, cysteines that are proposed to be poorly or not acylated could actually be excellent substrates for APTs. Since specific inhibitors of APTs are available, a control experiment in which claudins are purified in the presence of inhibitors is required and could actually add great value to this research.

.

Blaskovic, S., Blanc, M. and van der Goot, F. G. (2013). What does S-palmitoylation do to membrane proteins? *FEBS J* 280, 2766-74.

Dallavilla, T., Abrami, L., Sandoz, P. A., Savoglidis, G., Hatzimanikatis, V. and van der Goot, F. G. (2016). Model-Driven Understanding of Palmitoylation Dynamics: Regulated Acylation of the Endoplasmic Reticulum Chaperone Calnexin. *PLoS Comput Biol* 12, e1004774.

Gonzalez Montoro, A., Chumpen Ramirez, S., Quiroga, R. and Valdez Taubas, J. (2011). Specificity of transmembrane protein palmitoylation in yeast. *PLoS One* 6, e16969.

Lam, K. K., Davey, M., Sun, B., Roth, A. F., Davis, N. G. and Conibear, E. (2006). Palmitoylation by the DHHC protein Pfa4 regulates the ER exit of Chs3. *J Cell Biol* 174, 19-25.

Liang, X., Lu, Y., Neubert, T. A. and Resh, M. D. (2002). Mass spectrometric analysis of GAP-43/neuromodulin reveals the presence of a variety of fatty acylated species. *J Biol Chem* 277, 33032-40.

Ponimaskin, E. and Schmidt, M. F. (1995). Acylation of viral glycoproteins: structural requirements for palmitoylation of transmembrane proteins. *Biochem Soc Trans* 23, 565-8.

ten Brinke, A., Vaandrager, A. B., Haagsman, H. P., Ridder, A. N., van Golde, L. M. and Batenburg, J. J. (2002). Structural requirements for palmitoylation of surfactant protein C precursor. *Biochem J* 361, 663-71.

Yik, J. H. N. and Weigel, P. H. (2002). The Position of Cysteine Relative to the Transmembrane Domain Is Critical for Palmitoylation of H1, the Major Subunit of the Human Asialoglycoprotein Receptor. *J. Biol. Chem.* 277, 47305-47312.

Reviewer #2 (Remarks to the Author):

The manuscript by Rodenburg, et.al. explores protein S-palmitoylation on several membrane proteins (claudins, CD20, bacterial KscA and bacteriorhodopsin) by native mass spectrometry. This work is a technical breakthrough for studying protein S-palmitoylation, taking advantage of EMR Orbitrap technology pioneered in the Heck lab. While the technical feats accomplished in this manuscript are a milestone and will be extremely interesting to the protein lipidation community, the biological findings are somewhat over-reaching given the data presented. Clearly palmitoylation is not going to occur inside a transmembrane domain, and intracellular cysteines near the membrane surface are poised to undergo palmitoylation, either by enzymatic or non-enzymatic means. The authors argue that there should be no substrate specificity for palmitoylation, but presumably only spatial regulation through exposure to distinct palmitoyl transferases. If this is the case, then show it! The membrane proximal cysteines are poised at the membrane interface where palmitoyl-CoA is also present, which could readily exchange at these sites. While the majority of cysteine scanning mutants are palmitoylated, this data provides no evidence of whether this effect is enzyme-dependent. Is there literature precedence for distinct DHHC enzymes catalyzing claudin palmitoylation? Before the authors argument is believable, there should be some minimal exploration into the enzymatic role of palmitoylation of claudins. Do all zDHHC enzymes target claudins? How is claudin palmitoylation affected by 2-bromopalmitate? Is claudin palmitoylation enhanced by overexpression of specific DHHC enzymes? The cooperativity (while beyond my scope for evaluation) is also not surprising, although the quantitation is a unique approach to demonstrate the effect in a quantitative way. Overall, the data is quite interesting, but the discussion is should be open to more possible routes of regulation, both enzymatic and non-enzymatic. Furthermore, 2 mammalian proteins (claudins + CD20) does not make for a broad dataset for global characterizations. Clearly many proteins will have differential palmitoylation across membrane proximal cysteine residues. Again, the manuscript is technically a breakthrough, but should incorporate additional data to support their rather broad claims about DHHC enzymes and specificity, or narrow their claims.

Additional points:

1) Purified claudins were not assembled into tight junctions after treatment with C-CPE, which could change the palmitoylation dynamics quite a bit. Is there any way to look at native claudins without C-CPE? Are these effects an artifact of non-natural over-expression?

Reviewer #3 (Remarks to the Author):

The Rodenburg et al. manuscript takes a big step forward with both technological advancement and its impact on our understanding of transmembrane protein palmitoylation. With regard to technology, the authors take a new approach that uses MS to look at palmitoylation stoichiometry, which allows them to directly count the number of attached fats on a protein. This sounds trivial, but it is not. While the many new methods developed over the last ten years to facilitate palmitoylation detection, allow one to say if a protein is palmitoylated, they provide little information regarding stoichiometry. For particular palmitoylated cysteines, typically there is no way of distinguishing if a cysteine is used infrequently as an acceptor, say at a level of 1%, or more robustly, say at a 99% level. In the last year, this problem has been approached by a couple of other groups that have used ABE chemistry to substitute large compounds at palmitoylation sites, allowing discernible electrophoretic shifts to be detected in Western blots. The advantage of the Rodenburg approach is its directness – no substitution chemistry is required, with the investigators instead looking at the mass shift directly associated with each added palmitate. This technology, I expect, will be widely exploited in the future.

The authors focus here on the palmitoylated cysteines that map within the four claudin membrane-spanning domains. Such transmembrane domain palmitoylation is expected to change membrane protein affinity for membrane microdomains in that the saturated fats facilitate better packing with raft fats. Whole proteome analyses have identified palmitoylated cysteines in quite a large collection of transmembrane proteins, including many SNAREs, channels, receptors, etc. The stoichiometry at which these cysteines are utilized for palmitoylation for any of these proteins has remains largely unknown. CLDN3 has five candidate transmembrane cysteines and the authors convincingly show that 4 of these 5 are readily used for palmitoylation with high stoichiometry, so that the average CLDN3 purified from HEK membranes has 3 or 4 attached palmitates. As is typical, these transmembrane cysteines map to the predicted cytoplasmic leaflet of the membrane spanning domain, in proximity to the cytoplasm-membrane interface where they might be expected to access the DHHC enzymes responsible for attaching the fats. The authors go on to test whether new cysteines introduced into these portions of CLDN3 as well as into other, non-palmitoylated transmembrane proteins might also serve as acceptors and find this to be largely the case. These results yield two important conclusions. First, that palmitoylation is limited to the 8 angstroms of the transmembrane domain most proximal to the cytoplasm. Second, beyond mapping to this transmembrane space, little or no sequence specificity is required to direct these palmitoylations.

Overall, this is a very careful, thorough and thoughtful work that leads to a number of new, broadly significant insights. I particularly appreciated the included control experiments, e.g. the effects of expression level and freezer storage on palmitoylation stoichiometry. The suggestions below are aimed largely at improving and/or broadening the presentation.

1. Cooperativity. Not being a math wiz, I cannot speak with expertise on the methods used to reach the conclusion that cooperativity is at play in the palmitoylation of these multiple acceptors. However, my gross take is that such cooperativity is not wholly determinative. First, it is clear that the four cysteinyl acceptors behave grossly independently in that none of the single cysteine mutations substantially impact overall CLDN3 palmitoylation beyond reducing the average number of added palmitates from 4 to 3. My sense is that the cooperativity identified here is, in fact, subtle. The authors should be a bit clearer about this.

2. Perhaps the authors should say something more about the wider applicability of their new approach. Is this approach only practical for relatively small proteins? How pure must the protein be? etc

3. p 8,9. I found the discussion on the effect of transmembrane domain tilt on palmitoylation site usage a bit difficult. It was made clearer in the Discussion, particularly with the presentation of Fig. 4. This explanation, which presently first appears in the Results section, might be best reserved for the Discussion. One can simply note the surprising under-palmitoylation of Helix 3 sites and point the reader to the Discussion for further explanation.

4. Palmitoylation generally is thought to be somewhat promiscuous with regard to which acyl chains are used. This work provides one of the rare looks at the acyl chains that are used for "palmitoylation" and indeed, the 16C palmitate clearly is predominantly used. Perhaps this is worth a sentence or two of discussion.

5. p18, line 498. typo: tabacco → tobacco

Reviewers' comments:

Overall reply:

We thank the reviewers for their constructive critiques, which allowed us to present our data more clearly in the manuscript. Next to changes in response to their specific issues, we have adapted the title and abstract in line with their comments.

See changes in title and abstract manuscript.

Reviewer #1 (Remarks to the Author):

Using a novel MS technique combined with mathematical modelling and mutagenesis analyses the authors address a number of issues regarding palmitoylation of integral membrane proteins, particularly in cysteines that are close to the cytosolic border of transmembrane domains.

Overall, the work, which is technically challenging, has been performed meticulously and the conclusions are supported by the data.

An important problem encountered throughout the article is lack of reference and discussion of previous work in the field.

- Up to now, techniques to investigate the stoichiometry of s-acylation were not readily available and, in this respect, the paper represents a breakthrough. Nevertheless, stoichiometry of S-acylation has been achieved using alternative MS methods (Liang et al., 2002). This work should be cited and the improvements over these methods should be stated.

Reply 1.1:

Liang et al. rely on proteolytic digestion followed by MALDI-TOF MS to analyze palmitoylation in a peripheral membrane protein GAP43. To reduce background in the MS spectra resulting from many proteolytic digestion products, an engineered TEV recognition site was inserted in GAP43 used for proteolytic digestion. These analyses allow direct detection of palmitoylation and site-specific occupancy of acylation to be determined by relative MS signal, but they do not allow one to determine the overall stoichiometry of multiply acylated chains, especially as the sites of modification map to different proteolytic digestion products.

A sentence referencing (Liang, X. et al. *J. Biol. Chem.* **277**, 33032–33040 (2002)) and other methods (see reviewer 3 item A) were added to the introduction on page 4. The next sentence was slightly modified:

“In addition, protein engineering (Liang, X. et al. *J. Biol. Chem.* **277**, 33032–33040 (2002)) and replacement of palmitates by PEG have been employed to study protein palmitoylation (Percher, A. et al. *PNAS* **113**, 4302–4307 (2016), Yokoi, N. et al. *J. Neurosci.* **36**, 6431–6444 (2016)). However, the lack of methods to detect palmitoylation of native proteins has hindered detailed studies ...”

- The authors prove that cysteines that are deeper than 8A within the membrane, are not

palmitoylated, and this is of interest for the field. It would be useful if the authors could make a rough estimate, or discuss the 8Å distance correspondence, to an approximate position of a cysteine in a TMD, for membrane proteins in general. It is clear that the distance would depend on the tilt of a given TMD. Nevertheless, Assuming little to no tilt, up to which positions in the TMD can cysteines be palmitoylated?. This information, however approximate, would be very useful to the reader and it would allow comparison with existing data. Experimentally, it has been shown that cysteines in position 6 from the border toward the membrane are palmitoylated but cysteines that are located further are not (Ponimaskin and Schmidt, 1995).

Reply 1.2:

Given the pitch (1.5 Å) per residue in an alpha helix, 8 Å corresponds to 5 (5.3) residues, which would hold for a non-tilted helix.

The reviewer refers to (Ponimaskin, E. et al. *Biochem. Soc. Trans.* **23**, 565–568 (1995)); however, the depth of sites in the membrane is not addressed in this paper. This issue was addressed for palmitoylation of type II membrane proteins in (Schweizer, A. J. et al. *Biol. Chem.* **270**, 9638–9644 (1995), Brinke, ten, A. et al. *Biochemical Journal* 663–671 (2002), Yik, J. H. N. et al. *J. Biol. Chem.* **277**, 47305–47312 (2002), González Montoro, A. et al. *PLoS ONE* **6**, e16969 (2011)). In these papers, proteins were mutated moving the naturally occurring palmitoylation sites outwards to the cytosol (thus, not deeper into the membrane).

The additional papers were added to the references on page 3:

“Even though some motifs have been linked to protein palmitoylation, there is no strict generic, conserved sequence motif (Schweizer, A. J. et al. *Biol. Chem.* **270**, 9638–9644 (1995), Brinke, ten, A. et al. *Biochemical Journal* 663–671 (2002), Yik, J. H. N. et al. *J. Biol. Chem.* **277**, 47305–47312 (2002), González Montoro, A. et al. *PLoS ONE* **6**, e16969 (2011), Blanc, M. et al. *F1000Res* **4**, 261 (2015)). For membrane proteins, palmitoylated cysteines are generally located at the cytoplasmic side or in trans-membrane helices at the inner leaflet of the membrane (Schweizer, A. J. et al. *Biol. Chem.* **270**, 9638–9644 (1995), Brinke, ten, A. et al. *Biochemical Journal* 663–671 (2002), Yik, J. H. N. et al. *J. Biol. Chem.* **277**, 47305–47312 (2002), González Montoro, A. et al. *PLoS ONE* **6**, e16969 (2011)).”

We added the following phrase to the discussion on page 12:

“This distance corresponds to 5-6 residues for a helix perpendicular in the membrane.”

-The authors demonstrate conclusively that it is the position of the cysteines rather than any sequence requirement that is necessary for cysteine palmitoylation. The use of a mutated prokaryotic protein to study the sequence requirements for protein palmitoylation is particularly clever. However, the work reinforces by novel and elegant techniques, ideas that have already been proposed, most importantly by (ten Brinke et al., 2002; Yik and Weigel, 2002) (Citations to these two last articles are misplaced in the text). These articles should be properly cited and discussed. Other authors have also address this issues, perhaps less directly, particularly in the field of viral protein palmitoylation, and more recently by (Gonzalez Montoro et al., 2011)

Reply 1.3:

Referencing was addressed by the sentence added to the introduction in response to item 1.2 (see reply 1.2).

In addition, the following sentence was added to the discussion on page 13:

“These data are in line with observations on single-pass type-II membrane proteins that the local amino-acid sequence is not essential for palmitoylation (Schweizer, A. J. *et al. Biol. Chem.* **270**, 9638–9644 (1995), Brinke, ten, A. *et al. Biochemical Journal* 663–671 (2002), Yik, J. H. N. *et al. J. Biol. Chem.* **277**, 47305–47312 (2002), González Montoro, A. *et al. PLoS ONE* **6**, e16969 (2011)).”

-(Dallavilla et al., 2016) have addressed the issue of cooperativity in the palmitoylation of cysteines that are close to each other and to the TMD for Calnexin, and they propose a high degree of cooperativity. This should be cited and discussed. It was difficult to discern whether the authors find differences in the cooperativity of palmitoylation between cysteines that are close to each other in a TMD or that are in different TMDs. This could be an interesting point to address.

Reply 1.4:

Dallavilla et al. described a strict sequential pattern of palmitoylation (implying strict cooperativity) of two adjacent cysteines in type-I membrane protein calnexin. We observed stochastic palmitoylation with weak cooperativity for multi-spanning membrane proteins, independent of location of the palmitoylation site. This difference might be due to different PATs targeting the studied proteins. Calnexin is palmitoylated by DHH6 (Lakkaraju, A. K. *et al. EMBO J.* **31**, 1823–1835 (2012)). The PATs palmitoylating CD20 and claudins have not been identified yet, but since these proteins belong to a different class of membrane proteins it is possible that palmitoylation occurs via other PATs.

Changes in manuscript

Page 3, introduction, added:

“Pulse-chase data on calnexin suggested a preferred order to produce dual-palmitoylated species (Dallavilla, T. *et al. PLoS Comput. Biol.* **12**, e1004774 (2016)).”

Page 14, discussion, added:

“However, our observations of stochastic palmitoylations are in stark contrast to sequential palmitoylation reported for two neighboring cysteines in calnexin (Dallavilla, T. *et al. PLoS Comput. Biol.* **12**, e1004774 (2016)). Possibly, specific palmitoylation for calnexin by DHH6 (Lakkaraju, A. K. *et al. EMBO J.* **31**, 1823–1835 (2012)) and non-specific stochastic palmitoylations, as described here, are distinct cellular processes.”

Page 15, discussion, modified:

“Non-specific palmitoylation of multiple sites on membrane proteins does not follow a predefined sequential order”

-Palmitoylation is known to interfere with the subcellular localization of multispanning membrane proteins i.e. (Lam et al., 2006) reviewed in (Blaskovic et al., 2013). It is important that the subcellular localization of the mutant proteins is assessed. It is possible that mutants used in this study would be incorrectly localised and therefore inaccessible, or less accessible to its PAT. This would interfere with the interpretation of the proposed palmitoylation rates.

Reply 1.5

The reviewer raises an important point. Different subcellular localization of mutant and wild-type proteins could substantially influence palmitoylation rates. In our quantitative analysis, we used palmitoylation rates observed for single site mutants to predict palmitoylation distribution of wild-type claudin 3 and CD20. The accuracy of these predictions was very good, i.e. 96 and 98% accuracy, respectively. The accuracy of these predictions implies that palmitoylation rates of claudin and CD20 mutants do not differ from their native forms.

We modified and added the following sentences to emphasize this aspect (page 13):

“Analysis of proteins with multiple-palmitoylation sites and their single-site mutants showed that the observed distributions of palmitoylation isoforms can be predicted with high accuracy from the single-site palmitoylation rates (96 % and 98 % for Cld3 and CD20, respectively). This very high accuracy implies that the presented measurements are very precise and that no apparent effects are observed for potential differences in subcellular localization of mutants versus wild-type proteins.”

- An important aspect that has not been included in the arguments is de-palmitoylation. The profiles of palmitoylated proteins observed could very well be the result of a balance between acylation –deacylation. There are not many examples of integral membrane proteins that are actively de-palmitoylated (in cysteines close to their TMDs). Nevertheless, cysteines that are proposed to be poorly or not acylated could actually be excellent substrates for APTs. Since specific inhibitors of APTs are available, a control experiment in which claudins are purified in the presence of inhibitors is required and could actually add great value

Reply 1.6

Acylprotein thioesterase (APTs) found to-date are soluble lipid-anchored proteins. Consequently, the majority of thioesterase substrates are peripheral membrane proteins. In a recent paper, rapid depalmitoylation was observed for peripheral membrane protein PSD-95 (Yokoi, N. et al. *Enzymes. J. Neurosci.* **36**, 6431–6444 (2016)). Using the same experimental setup, these authors observe no depalmitoylation for membrane proteins GluA1, GluN2A and mGluR5. β_1 AR contains two palmitoylation sites. Zuckerman et al. show the site in the trans-membrane region is not depalmitoylated, whereas the second site, located on the C-terminal flexible tail, is rapidly depalmitoylated (Zuckerman, D. M. et al. *J. Biol. Chem.* **286**, 19014–19023 (2011)). Moreover, recent crystal structures of APT1 and APT2 suggested that substrates with reduced membrane partitioning, such as mono-palmitoylated proteins, are the primary substrate for APT enzymes (Won, S. J. et al. *ACS Chem. Biol.* **11**, 3374–3382 (2016)).

The following text was added to the discussion on page 13/14:

“Depalmitoylation of membrane proteins has been observed for sites on intracellular loops (Tian, L. et al. *J. Biol. Chem.* **287**, 14718–14725 (2012)) and flexible termini (Zuckerman, D. M. et al. *J. Biol. Chem.* **286**, 19014–19023 (2011)). Our data is consistent with stochastic palmitoylation both with and without potential stochastic depalmitoylation.”

Blaskovic, S., Blanc, M. and van der Goot, F. G. (2013). What does S-palmitoylation do to membrane proteins? FEBS J 280, 2766-74.

Dallavilla, T., Abrami, L., Sandoz, P. A., Savoglidis, G., Hatzimanikatis, V. and van der Goot, F. G. (2016). Model-Driven Understanding of Palmitoylation Dynamics: Regulated Acylation of the Endoplasmic Reticulum Chaperone Calnexin. PLoS Comput Biol 12, e1004774.

Gonzalez Montoro, A., Chumpen Ramirez, S., Quiroga, R. and Valdez Taubas, J. (2011). Specificity of transmembrane protein palmitoylation in yeast. PLoS One 6, e16969.

Lam, K. K., Davey, M., Sun, B., Roth, A. F., Davis, N. G. and Conibear, E. (2006). Palmitoylation by the DHHC protein Pfa4 regulates the ER exit of Chs3. J Cell Biol 174, 19-25.

Liang, X., Lu, Y., Neubert, T. A. and Resh, M. D. (2002). Mass spectrometric analysis of GAP-43/neuromodulin reveals the presence of a variety of fatty acylated species. J Biol Chem 277, 33032-40.

Ponimaskin, E. and Schmidt, M. F. (1995). Acylation of viral glycoproteins: structural requirements for palmitoylation of transmembrane proteins. Biochem Soc Trans 23, 565-8.

ten Brinke, A., Vaandrager, A. B., Haagsman, H. P., Ridder, A. N., van Golde, L. M. and Batenburg, J. J. (2002). Structural requirements for palmitoylation of surfactant protein C precursor. Biochem J 361, 663-71.

Yik, J. H. N. and Weigel, P. H. (2002). The Position of Cysteine Relative to the Transmembrane Domain Is Critical for Palmitoylation of H1, the Major Subunit of the Human Asialoglycoprotein Receptor. J. Biol. Chem. 277, 47305-47312.

Reviewer #2 (Remarks to the Author):

The manuscript by Rodenburg, et.al. explores protein S-palmitoylation on several membrane proteins (claudins, CD20, bacterial KscA and bacteriorhodopsin) by native mass spectrometry. This work is a technical breakthrough for studying protein S-palmitoylation, taking advantage of EMR Orbitrap technology pioneered in the Heck lab. While the technical feats accomplished in this manuscript are a milestone and will be extremely interesting to the protein lipidation community, the biological findings are somewhat over-reaching given the data presented. Clearly palmitoylation is not going to occur inside a transmembrane domain, and intracellular cysteines near the membrane surface are poised to undergo palmitoylation, either by enzymatic or non-enzymatic means. The authors argue that there should be no substrate specificity for palmitoylation, but presumably only spatial regulation through exposure to distinct palmitoyl transferases. If this is the case, then show it! The membrane

proximal cysteines are poised at the membrane interface where palmitoyl-CoA is also present, which could readily exchange at these sites. While the majority of cysteine scanning mutants are palmitoylated, this data provides no evidence of whether this effect is enzyme-dependent. Is there literature precedence for distinct DHHC enzymes catalyzing claudin palmitoylation? Before the authors argument is believable, there should be some minimal exploration into the enzymatic role of palmitoylation of claudins. Do all zDHHC enzymes target claudins?

Reply 2.1

The reviewer is skeptical about enzymatic palmitoylation. Even though palmitoylation may occur spontaneously *in vitro* as shown by (Duncan, J. A. et al. *J. Biol. Chem.* **271**, 23594–23600 (1996), Dietrich, L. E. P. et al. *EMBO J.* **23**, 45–53 (2004)). Roth et al. have shown that *in vivo* palmitoylation is DHHC mediated (Roth, A. F. et al. *Cell* **125**, 1003–1013 (2006)). Using DHHC deficient yeast strains, it was shown that DHHC proteins were involved in the palmitoylation of 29 proteins out of a panel of 30 palmitoylated proteins. The one outlier had its palmitoyl chain buried inside the protein core. Specifically, *in vivo* Vac8 palmitoylation was shown to be fully PAT dependent, whereas, *in vitro*, it can be palmitoylated in a PAT independent manner (Roth, A. F. et al. *Cell* **125**, 1003–1013 (2006)). Moreover, *in vivo*, non-enzymatic palmitoylation is inhibited by highly abundant cytosolic acyl-CoA binding protein (Dunphy, J. T. et al. *Biochim. Biophys. Acta* **1485**, 185–198 (2000)). Therefore, palmitoylation is widely regarded to be enzymatically driven *in vivo*, see e.g. (Blaskovic, S. et al. *FEBS J.* **280**, 2766–2774 (2013), Resh, M. D. *Progress in Lipid Research* **63**, 120–131 (2016)) in addition to (Salaun, C. et al. *J. Cell Biol.* **191**, 1229–1238 (2010)), which is the first reference in the manuscript. Moreover, whether or not some small fraction of palmitoylation could be accounted for by non-enzymatic means in the current experiments, the implications of our findings for biochemistry and protein engineering remain the same, *i.e.* that one should expect to see a palmitoylation if a protein presents a free cysteine within 8 Angstrom of the cytosol in the plasma membrane.

Sentences added to discussion:

“*In vivo* palmitoylation is expected to be enzymatic, and our results indicate that it is not sequence specific, but rather governed by the accessibility of the free cysteine. Our data do not exclude a non-enzymatic process, however the implication remains identical, *i.e.* palmitoylation of free, accessible cysteines occur up to 8 Å into the membrane on the cytosolic side.”

How is claudin palmitoylation affected by 2-bromopalmitate?

Reply 2.2

The influence of claudin palmitoylation by 2-bromopalmitate is unknown. 2-Bromopalmitate is a very promiscuous palmitoylation inhibitor (Davda, D. et al. *ACS Chem. Biol.* **8**, 1912–1917 (2013)), therefore it is questionable whether it can be used to assess whether claudin palmitoylation is an enzymatic process. See also reply to previous item.

Is claudin palmitoylation enhanced by overexpression of specific DHHC enzymes?

Reply 2.3

The human genome contains 23 DHHC enzymes (Mitchell, D. A. et al. *J. Lipid Res.* **47**, 1118–1127 (2006)), investigating whether co-expression with every DHHC enzyme enhances claudin palmitoylation is beyond the scope of this study.

No changes made to manuscript.

The cooperativity (while beyond my scope for evaluation) is also not surprising, although the quantitation is a unique approach to demonstrate the effect in a quantitative way. Overall, the data is quite interesting, but the discussion is should be open to more possible routes of regulation, both enzymatic and non-enzymatic. Furthermore, 2 mammalian proteins (claudins + CD20) does not make for a broad dataset for global characterizations. Clearly many proteins will have differential palmitoylation across membrane proximal cysteine residues. Again, the manuscript is technically a breakthrough, but should incorporate additional data to support their rather broad claims about DHHC enzymes and specificity, or narrow their claims.

Reply 2.4

In the final paragraph we narrow our observations to be due to a “generic, non-specific process of membrane palmitoylation” only.

Changes in text, we added “non-specific” twice in final paragraph on page 15:
“...our study revealed a generic, non-specific process of membrane palmitoylation”.
“Non-specific palmitoylation of multiple sites...”

Additional points:

1) Purified claudins were not assembled into tight junctions after treatment with C-CPE, which could change the palmitoylation dynamics quite a bit. Is there any way to look at native claudins without C-CPE? Are these effects an artifact of non-natural over-expression?

Reply 2.5

We determined mass spectra for Cld3 without addition of C-CPE, which are very similar to Cld3 with C-CPE. The effect of over-expression on palmitoylation assessed by lowering expression levels and was found to be negligible (Supplementary figure 3a).

Supplementary figure 1 was adapted to accommodate the apo-Cld3 mass spectrum.

The following sentence was added to the results section on page 5:

“Cld3 purified without C-CPE addition revealed a very similar mass spectra, indicating that C-CPE does not substantially alter palmitoylation of Cld3 (Supplementary figure 1b).”

and accordingly to materials and methods, page 21:

“...(no C-CPE was added to obtain Cld3 without C-CPE)...”

Reviewer #3 (Remarks to the Author):

The Rodenburg et al. manuscript takes a big step forward with both technological advancement

and its impact on our understanding of transmembrane protein palmitoylation. With regard to technology, the authors take a new approach that uses MS to look at palmitoylation stoichiometry, which allows them to directly count the number of attached fats on a protein. This sounds trivial, but it is not. While the many new methods developed over the last ten years to facilitate palmitoylation detection, allow one to say if a protein is palmitoylated, they provide little information regarding stoichiometry. For particular palmitoylated cysteines, typically there is no way of distinguishing if a cysteine is used infrequently as an acceptor, say at a level of 1%, or more robustly, say at a 99% level. In the last year, this problem has been approached by a couple of other groups that have used ABE chemistry to substitute large compounds at palmitoylation sites, allowing discernible electrophoretic shifts to be detected in Western blots. The advantage of the Rodenburg approach is its directness – no substitution chemistry is required, with the investigators instead looking at the mass shift directly associated with each added palmitate. This technology, I expect, will be widely exploited in the future.

Reply 3-A

A sentence referencing the use of ABE chemistry to detect palmitoylation stoichiometry was added to the introduction on page 4:

*“In addition, protein engineering (Liang, X. et al. *J. Biol. Chem.* **277**, 33032–33040 (2002)) and replacement of palmitates by PEG have been employed to study protein palmitoylation (Percher, A. et al. *PNAS* **113**, 4302–4307 (2016), Yokoi, N. et al. *J. Neurosci.* **36**, 6431–6444 (2016)). However, the lack of methods to detect palmitoylation of native proteins has hindered detailed studies...”*

The authors focus here on the palmitoylated cysteines that map within the four claudin membrane-spanning domains. Such transmembrane domain palmitoylation is expected to change membrane protein affinity for membrane microdomains in that the saturated fats facilitate better packing with raft fats. Whole proteome analyses have identified palmitoylated cysteines in quite a large collection of transmembrane proteins, including many SNAREs, channels, receptors, etc. The stoichiometry at which these cysteines are utilized for palmitoylation for any of these proteins has remains largely unknown. CLDN3 has five candidate transmembrane cysteines and the authors convincingly show that 4 of these 5 are readily used for palmitoylation with high stoichiometry, so that the average CLDN3 purified from HEK membranes has 3 or 4 attached palmitates. As is typical, these transmembrane cysteines map to the predicted cytoplasmic leaflet of the membrane spanning domain, in proximity to the cytoplasm-membrane interface where they might be expected to access the DHHC enzymes responsible for attaching the fats. The authors go on to test whether new cysteines introduced into these portions of CLDN3 as well as into other, non-palmitoylated transmembrane proteins might also serve as acceptors and find this to be largely the case. These results yield two important conclusions. First, that palmitoylation is limited to the 8 angstroms of the transmembrane domain most proximal to the cytoplasm. Second, beyond mapping to this transmembrane space, little or no sequence specificity is required to direct these palmitoylations.

Overall, this is a very careful, thorough and thoughtful work that leads to a number of new, broadly significant insights. I particularly appreciated the included control experiments, e.g. the effects of expression level and freezer storage on palmitoylation stoichiometry. The suggestions below are aimed largely at improving and/or broadening the presentation.

1. Cooperativity. Not being a math wiz, I cannot speak with expertise on the methods used to reach the conclusion that cooperativity at play in the palmitoylation of these multiple acceptors. However, my gross take is that such cooperativity is not wholly determinative. First, it is clear that the four cysteinyl acceptors behave grossly independently in that none of the single cysteine mutations substantially impact overall CLDN3 palmitoylation beyond reducing the average number of added palmitates from 4 to 3. My sense is that the cooperativity identified here is, in fact, subtle. The authors should be a bit clearer about this.

Reply 3.1

Indeed the observed cooperativity is weak (1.25, meaning that following a previous palmitoylation of a protein molecule the chance for a subsequent palmitoylation is enhanced by 25%). We attribute this effect to enhanced presence of PATs in the vicinity of its target after previous palmitoylation.

This is described on page 13/14:

“The process yielding multiple palmitoylations is weakly cooperative, i.e. the chance of subsequent palmitoylation increases by ca. 25%. This weak cooperativity may be explained by a locally increased level of PAT-enzymes present due to prior palmitoylation of the multiply palmitoylated membrane protein.”

No changes made to manuscript.

2. Perhaps the authors should say something more about the wider applicability of their new approach. Is this approach only practical for relatively small proteins? How pure must the protein be? etc

Reply 3.2

We anticipate its applicability is similar to the mass spectrometric detection of unpalmitoylated membrane proteins as described in Hopper et al. and other papers from the Robinson, Sobott and Ashcroft labs, which have published native MS data on detergent and amphipol solubilized membrane protein complexes up to ~500 kDa in mass (Hopper, J. T. S. et al. *Nature Methods* **10**, 1206–1208 (2013)). Purity requirements vary widely and are primarily limited by how close the analyte and background signals are in mass-to-charge ratio. For well-separated signals, purity is trivial because MS is a technique with a tremendous dynamic range that spans orders of magnitudes in intensity. However, low-level contaminants that just so happen to appear at the same mass-to-charge ratio may already interfere with the relative accuracy of quantitation, though this is also mitigated by quantitation based on multiple charge states. To illustrate, the referee may consider the SDS-PAGE gels of claudin presented in the supplementary information, which indicate that claudin is less than 50% pure by stain density on gel, whereas the

preparations of KcsA and bacteriorhodopsin required a higher purity.

We added reference to Hopper et al. on page 12.

“When these technical requirements are fulfilled (Hopper, J. T. S. et al. *Nature Methods* **10**, 1206–1208 (2013)), native MS provides an approach...”

3. p 8,9. I found the discussion on the effect of transmembrane domain tilt on palmitoylation site usage a bit difficult. It was made clearer in the Discussion, particularly with the presentation of Fig. 4. This explanation, which presently first appears in the Results section, might be best reserved for the Discussion. One can simply note the surprising under-palmitoylation of Helix 3 sites and point the reader to the Discussion for further explanation.

Reply 3.3

We removed the geometric description and refer now directly to the resulting effect in the structure as shown in Supplementary Fig. 7b. This clarifies the text on page 9.

“For helix 3, the palmitoylation sites lie on a lipid-exposed surface, which forms an acute angle with the bilayer plane (Supplementary Fig. 7b).”

4. Palmitoylation generally is thought to be somewhat promiscuous with regard to which acyl chains are used. This work provides one of the rare looks at the acyl chains that are used for “palmitoylation” and indeed, the 16C palmitate clearly is predominantly used. Perhaps this is worth a sentence or two of discussion.

Reply 3.4

We rephrased the text in the discussion on page 14 to stress this point:

“Acyl chains of different lengths may be attached to cysteines in membrane proteins (Resh, M. D. *Progress in Lipid Research* **63**, 120–131 (2016)). However, we detected palmitate modifications only.”

5. p18, line 498. typo: tabacco → tobacco

Reply 3.5

The typo is corrected.

REVIEWERS' COMMENTS:

Reviewer #1 (Remarks to the Author):

The authors have addressed most of the concerns raised in my review. However, some issues remain

Reply 1.5

The reviewer raises an important point. Different subcellular localization of mutant and wildtype proteins could substantially influence palmitoylation rates. In our quantitative analysis, we used palmitoylation rates observed for single site mutants to predict palmitoylation distribution of wild-type claudin 3 and CD20. The accuracy of these predictions was very good, i.e. 96 and 98% accuracy, respectively. The accuracy of these predictions implies that palmitoylation rates of claudin and CD20 mutants do not differ from their native forms. We modified and added the following sentences to emphasize this aspect (page 13): "Analysis of proteins with multiple-palmitoylation sites and their single-site mutants showed that the observed distributions of palmitoylation isoforms can be predicted with high accuracy from the single-site palmitoylation rates (96 % and 98 % for Cld3 and CD20, respectively). This very high accuracy implies that the presented measurements are very precise and that no apparent effects are observed for potential differences in subcellular localization of mutants versus wild-type proteins."

-While I am satisfied with the explanation, the newly added sentences should be corrected and made more specific to Cld3 and CD20, the proteins addressed in this study. The fact that they do not observe differences in the palmitoylation that may be ascribed to changes in the subcellular localization of the mutants is not necessarily a general rule.

Reply 1.6

Acylprotein thioesterase (APTs) found to-date are soluble lipid-anchored proteins. Consequently, the majority of thioesterase substrates are peripheral membrane proteins. In a recent paper, rapid depalmitoylation was observed for peripheral membrane protein PSD-95 (Yokoi, N. et al. *Enzymes. J. Neurosci.* 36, 6431–6444 (2016)). Using the same experimental setup, these authors observe no depalmitoylation for membrane proteins GluA1, GluN2A and mGluR5. β 1AR contains two palmitoylation sites. Zuckerman et al. show the site in the transmembrane region is not depalmitoylated, whereas the second site, located on the C-terminal flexible tail, is rapidly depalmitoylated (Zuckerman, D. M. et al. *J. Biol. Chem.* 286, 19014–19023 (2011)). Moreover, recent crystal structures of APT1 and APT2 suggested that substrates with reduced membrane partitioning, such as mono-palmitoylated proteins, are the primary substrate for APT enzymes (Won, S. J. et al. *ACS Chem. Biol.* 11, 3374–3382 (2016)).

.

The following text was added to the discussion on page 13/14:

"Depalmitoylation of membrane proteins has been observed for sites on intracellular loops (Tian, L. et al. *J. Biol. Chem.* 287, 14718–14725 (2012)) and flexible termini (Zuckerman, D. M. et al. *J. Biol. Chem.* 286, 19014–19023 (2011)). Our data is consistent with stochastic palmitoylation both with and without potential stochastic depalmitoylation

The authors chose not to carry out the experiments I suggested on the basis that there is no evidence of palmitoylated cysteines close to TMDs that are substrates for APTs. I mentioned this in my review and these concepts were reinforced by the authors' rebuttal. Nevertheless, The field of protein de-palmitoylation is very young, only a handful of proteins have been examined, and do

not think anyone in the field can confidently say that this will always be the case. The experiment is interesting and simple enough and I believe the authors should have done it.

-Regarding the issue of enzymatic vs non-enzymatic palmitoylation raised by another reviewer, for cysteines that are at the border of a TMD, it was pretty much settled by the characterization of Swf1, the PAT that modifies this type of cysteines in yeast (Valdez Taubas and Pelham, 2005) and see review by Mitchel et al 2006, JLR) . The reference to Roth et al 2006 is also relevant as suggested by the authors.

Reviewer #2 (Remarks to the Author):

After re-reading the paper, I think the manuscript should further qualify their results more directly, and mention that their results are valid for these specific proteins, and may not be generalizable across all protein targets. Since the authors do not wish to explore DHHC dependence, it does leave some questions about specificity unanswered, making it rather over-reaching to make broad statements about selectivity when they have no enzyme-substrate relationship data. This is particularly important for the abstract and final paragraph, where their statements are somewhat over-reaching. It is appreciated that they qualified their data in the second to last paragraph of the revision, but there should be more careful qualifiers when making broad statements. Clearly they want to make a big statement about palmitoylation near membrane helices, but they need to be somewhat careful, especially in proteins purified from HEK cells that are not polarized and lack typically epithelial organization. As before, the manuscript should be accepted minor minor text revisions (and without further review).

Reviewer #3 (Remarks to the Author):

I am satisfied by the changes made to this revised manuscript. Overall, I remain quite positive about this manuscript. I think that these results will spark wide interest.

REVIEWERS' COMMENTS:

Reviewer #1 (Remarks to the Author):

The authors have addressed most of the concerns raised in my review. However, some issues remain

Reviewer comment 1.1

Reply 1.5

The reviewer raises an important point. Different subcellular localization of mutant and wildtype proteins could substantially influence palmitoylation rates. In our quantitative analysis, we used palmitoylation rates observed for single site mutants to predict palmitoylation distribution of wild-type claudin 3 and CD20. The accuracy of these predictions was very good, i.e. 96 and 98% accuracy, respectively. The accuracy of these predictions implies that palmitoylation rates of claudin and CD20 mutants do not differ from their native forms. We modified and added the following sentences to emphasize this aspect (page 13): "Analysis of proteins with multiple-palmitoylation sites and their single-site mutants showed that the observed distributions of palmitoylation isoforms can be predicted with high accuracy from the single-site palmitoylation rates (96 % and 98 % for Cld3 and CD20, respectively). This very high accuracy implies that the presented measurements are very precise and that no apparent effects are observed for potential differences in subcellular localization of mutants versus wild-type proteins."

-While I am satisfied with the explanation, the newly added sentences should be corrected and made more specific to Cld3 and CD20, the proteins addressed in this study. The fact that they do not observe differences in the palmitoylation that may be ascribed to changes in the subcellular localization of the mutants is not necessarily a general rule.

Reply 1.1

We agree with the reviewer and therefore clarified the statement to specifically address the studied proteins:

"Analysis of proteins with multiple-palmitoylation sites and their single-site mutants showed that the observed distributions of palmitoylation isoforms can be predicted with high accuracy from the single-site palmitoylation rates (96 % and 98 % for Cld3 and CD20, respectively). The accuracy of this prediction implies that there are no potential differences in subcellular localization of mutants versus wild-type Cld3 and CD20 that affect the overall accuracy of the palmitoylation measurement."

We additionally made some changes regarding general applicability of the proposed model in manuscript (see replies 2.1 and 2.2).

Reviewer comment 1.2

Reply 1.6

Acylprotein thioesterase (APTs) found to-date are soluble lipid-anchored proteins. Consequently, the majority of thioesterase substrates are peripheral membrane proteins. In a recent paper, rapid depalmitoylation was observed for peripheral membrane protein PSD-95 (Yokoi, N. et al. *Enzymes. J. Neurosci.* 36, 6431–6444 (2016)). Using the same experimental setup, these authors observe no depalmitoylation for membrane proteins GluA1, GluN2A and mGluR5. β 1AR contains two palmitoylation sites. Zuckerman et al. show the site in the transmembrane region is not depalmitoylated, whereas the second site, located on the C-terminal flexible tail, is rapidly depalmitoylated (Zuckerman, D. M. et al. *J. Biol. Chem.* 286, 19014–19023 (2011)). Moreover, recent crystal structures of APT1 and APT2 suggested that substrates with reduced membrane partitioning, such as mono-palmitoylated proteins, are the primary substrate for APT enzymes (Won, S. J. et al. *ACS Chem. Biol.* 11, 3374–3382 (2016)).

The following text was added to the discussion on page 13/14:

“Depalmitoylation of membrane proteins has been observed for sites on intracellular loops (Tian, L. et al. *J. Biol. Chem.* 287, 14718–14725 (2012)) and flexible termini (Zuckerman, D. M. et al. *J. Biol. Chem.* 286, 19014–19023 (2011)). Our data is consistent with stochastic palmitoylation both with and without potential stochastic depalmitoylation

The authors chose not to carry out the experiments I suggested on the basis that there is no evidence of palmitoylated cysteines close to TMDs that are substrates for APTs. I mentioned this in my review and these concepts were reinforced by the authors’ rebuttal. Nevertheless, The field of protein depalmitoylation is very young, only a handful of proteins have been examined, and do not think anyone in the field can confidently say that this will always be the case.

The experiment is interesting and simple enough and I believe the authors should have done it.

Reply 1.2

We agree with the reviewer that we cannot confidently exclude depalmitoylation occurring in our experiments. Therefore, we made mention of this in the discussion:

“Depalmitoylation of membrane proteins has been observed for sites on intracellular loops⁵⁶ and flexible termini⁵⁷, not for sites in and near transmembrane helices such as those studied here. Although we cannot exclude depalmitoylation occurring, our data is consistent with stochastic palmitoylation both with and without potential stochastic depalmitoylation.”

Also, we acknowledge the importance of depalmitoylation in the concluding sentence of the manuscript.

Reviewer comment 1.3

-Regarding the issue of enzymatic vs non-enzymatic palmitoylation raised by another reviewer, for

cysteines that are at the border of a TMD, it was pretty much settled by the characterization of Swf1, the PAT that modifies this type of cysteines in yeast (Valdez Taubas and Pelham, 2005) and see review by Mitchel et al 2006, JLR). The reference to Roth et al 2006 is also relevant as suggested by the authors.

Reply 1.3

We thank the reviewer for suggesting these additional references and we added them to the introduction:

“Palmitoylation is mediated by membrane embedded palmitoyl-acyl transferases (PAT) that contain a palmitoylated cysteine-rich domain with a conserved Asp-His-His-Cys (DHHC) motif, which is required for the palmitoylation activity¹⁴⁻¹⁸.”

Reviewer #2 (Remarks to the Author):

Reviewer comment 2.1

After re-reading the paper, I think the manuscript should further qualify their results more directly, and mention that their results are valid for these specific proteins, and may not be generalizable across all protein targets. Since the authors do not wish to explore DHHC dependence, it does leave some questions about specificity unanswered, making it rather over-reaching to make broad statements about selectivity when they have no enzyme-substrate relationship data. This is particularly important for the abstract and final paragraph, where their statements are somewhat over-reaching. It is appreciated that they qualified their data in the second to last paragraph of the revision, but there should be more careful qualifiers when making broad statements.

Reply 2.1

In line with the comments of the reviewer and other reviewers we thoroughly went through the manuscript and made changes to statements on general applicability of the proposed model.

In the abstract:

“These results suggest a generic, stochastic membrane-protein palmitoylation process that is determined by the accessibility of palmitoyl-acyl transferases to cysteines on membrane-embedded proteins, and not by a preferred substrate-sequence motif.”

In the introduction:

“Based on our data we propose a model in which palmitoylation of cysteines is not dependent on a local sequence motif. In these cases, modification of cysteines is rather determined by membrane-protein structure, in that it occurs generally on accessible residues within a depth of 8 Å of the lipid bilayer. Moreover, palmitoylation at multiple sites on Cld3 and CD20 is a stochastic but cooperative process. Collectively, these results provide insights into a membrane-protein palmitoylation process that is independent of a substrate-sequence motif.”

In the results:

“Since no cysteine palmitoylation occurs in prokaryotes, the ability to palmitoylate these prokaryotic proteins indicates strongly suggests that a single exposed cysteine at the right depth in the membrane is the sole requirement to induce palmitoylation.”

“Palmitoylation of Cld3 and CD20 is a stochastic and cooperative process”

In the discussion:

“Our native MS analysis of cysteine palmitoylation revealed that cysteines on Cld3 situated maximally 8 Å into the inner-membrane leaflet and exposed to the lipid environment become palmitoylated.”

“Analysis of these proteins with multiple-palmitoylation sites and their single-site mutants showed that the observed distributions of palmitoylation isoforms can be predicted with high accuracy from the single-site palmitoylation rates (96 % and 98 % for Cld3 and CD20, respectively).”

Multiple changes were made in the last paragraph:

“In conclusion, in our study we propose a model for revealed a the generic, non-specific process of membrane-proteins palmitoylation. In contrast to most other protein post-translational modifications, generic our model suggests that membrane-protein palmitoylation is not determined by a specific sequence motif, but merely by the accessibility of the substrate cysteine to the catalytic center of the PAT enzymes. Non-specific palmitoylation of multiple sites on membrane proteins does not follow a predefined sequential order, but is rather determined by a stochastic process yielding a distribution of palmitoylated isoforms. Even though this process is stochastic, palmitoylation of multiple sites on a membrane protein also exhibits a weak cooperativity. The rate of subsequent palmitoylations is slightly increased, because prior palmitoylation increases the likelihood of PAT enzymes in the vicinity of the substrate protein. The process of palmitoylation and depalmitoylation is both dynamic and highly complex and therefore requires both system wide and protein specific studies. The model proposed here provides detailed insight in the palmitoylation specificity of a set of proteins and suggests the existence of a general mechanism for palmitoylation. Future experiments testing the applicability of this model on proteins in their native environment are required to unravel the workings of dynamic palmitoylation and depalmitoylation.”

Reviewer comment 2.2

Clearly they want to make a big statement about palmitoylation near membrane helices, but they need to be somewhat careful, especially in proteins purified from HEK cells that are not polarized and lack typically epithelial organization. As before, the manuscript should be accepted minor minor text revisions (and without further review).

Reply 2.2

We agree with the reviewer that studies of protein palmitoylation require proteins to be investigated in their native or at least close to native environment. Therefore, we modified the concluding sentence of the manuscript:

“Future experiments testing the applicability of this model on proteins in their native environment are required to unravel the workings of dynamic palmitoylation and depalmitoylation.”

Reviewer #3 (Remarks to the Author):

I am satisfied by the changes made to this revised manuscript. Overall, I remain quite positive about this manuscript. I think that these results will spark wide interest.

Further changes made to improve clarity:

In the introduction:

“Even though some motifs have been linked to protein palmitoylation, there is no strictly conserved sequence motif²³⁻²⁷.”

In the discussion:

“Moreover, palmitoylation of cysteines introduced in prokaryotic membrane proteins bacteriorhodopsin and KcsA expressed in HEK cells demonstrates that palmitoylation of these proteins does not depend on recognition of a specific sequence or structural motif. These data are in line with observations on single-pass type-II membrane proteins that the specific local amino-acid sequence is not essential for palmitoylation²³⁻²⁶. In vivo palmitoylation is expected to be enzymatic¹⁶⁻¹⁸. Although our data do not exclude a non-enzymatic process, the implications remain identical, i.e. palmitoylation of free, accessible cysteines may occur up to 8 Å into the membrane on the cytosolic side and palmitoylation may occur independent of a sequence motif, but is governed by the accessibility of the free cysteine.”